# A Structural Equation Model Demonstrating the Relationship between Food Safety Background, Knowledge, Attitudes and Behaviour among Swedish Students

**DOI:** 10.3390/foods11111595

**Published:** 2022-05-28

**Authors:** Ingela Marklinder, Gustav Eskhult, Roger Ahlgren, Anna Blücher, Stina-Mina Ehn Börjesson, Madeleine Moazzami, Jenny Schelin, Marie-Louise Danielsson-Tham

**Affiliations:** 1Department of Food Studies, Nutrition and Dietetics, Uppsala University, 751 22 Uppsala, Sweden; 2Department of Statistics, Uppsala University, 751 20 Uppsala, Sweden; gustaveskhult@hotmail.com; 3Department of Food, Nutrition and Culinary Science, Umeå University, 901 87 Umeå, Sweden; roger.ahlgren2@gmail.com; 4Faculty of Health and Life Sciences, Linnaeus University, 430 38 Kalmar, Sweden; anna.blucher@lnu.se; 5Faculty of Natural Science, Kristianstad University, 291 88 Kristianstad, Sweden; stina-mina.ehn_borjesson@hkr.se; 6Department of Biomedical Sciences and Veterinary Public Health, Swedish University of Agricultural Sciences, 750 07 Uppsala, Sweden; madeleine.moazzami@slu.se; 7Division of Applied Microbiology, Department of Chemistry, Lund University, 221 00 Lund, Sweden; jenny.schelin@tmb.lth.se; 8Department of Meal Science and Culinary Arts, Örebro University, 701 82 Örebro, Sweden; marie-louise.danielsson-tham@oru.se

**Keywords:** university students, food safety, attitudes, structural equation model, food safety education

## Abstract

Traditionally, food safety knowledge has been seen as a factor in improving food safety behaviour. However, the relationship between knowledge and behavior is complex. The aim of the present study was to investigate self-reported data from 408 university students regarding food safety background, knowledge, attitudes, and behaviour using Structural Equation Model (SEM) to examine the influence of different factors on food safety behaviour. The SEM was applied to four factors derived from the data: Background, Knowledge, Attitude and Behaviour. The novelty of this current investigation is the inclusion of the Background factor (genus; experience of cooking and handling different food items; experience of a food safety education course; the foremost sources of food safety knowledge). The factors were constructed from variables with sufficient factor loadings and set up in a predetermined structure confirmed to be valid in previous studies. The results, demonstrated as regression coefficients between factors, confirm that the Background factor strongly influenced Knowledge (0.842). The Knowledge factor, in turn, strongly affected Attitude (0.605), while it did not directly affect Behaviour (0.301) in the same way as Attitude. Attitude had a stronger influence on Behaviour (0.438) than Knowledge. Thus, the Attitude factor seemed to play a mediating role between Knowledge and Behaviour. This indicates that students´ attitudes towards the importance of food safety may have an impact on their food safety behavior, which should have implications for the development of food safety education. This warrants further investigation and practical development.

## 1. Introduction

Traditionally, food safety knowledge has been seen as an important factor in improving food safety behaviour, and a substantial amount of information regarding consumer knowledge and self-reported practices has been reviewed [1,2,3,4,5]. However, there is no doubt that a complex relationship exists between knowledge and behaviour [6,7].

Mullan et al. [8] criticised interventions targeting food safety based on knowledge. Applying the Theory of Planned Behaviour (TPB) model, knowledge was demonstrated to be a necessary but not a sufficient condition for particular behaviours, and when other factors, such as norms and perceived control, were taken into accounts in the analysis, the knowledge factor was not the sole predictor of behaviour [8]. Food safety practices refers to handling in terms of cleaning, cold food storage, cooking, chilling, hand-washing, and avoidance of cross-contamination. The combination of Knowledge, Attitude, and Practice has been termed KAP [6,9,10]. Da Cunha [11] highlighted several caveats of the KAP approach related to difficulties translating knowledge into practices. Human beings are complex and factors such as optimistic bias, lack of motivation or inadequate infrastructure may hamper optimal behaviour [8,11]. Furthermore, risk perception is a complicated concept; people are generally more likely to believe that they will win the highest profit rather than that they will suffer from something negative, such as, for instance, food poisoning. This “optimistic bias” or “risk related optimism”, defined by Weinstein [12], may prevent consumers from absorbing information and following rules.

Actually, attitudes have been studied in relation to behaviour for decades. Ajzen [13] proposed that an individual’s attitude towards any object is a function of the strength of his or her personal beliefs about the object and the evaluative aspect of those beliefs. If people believe that a certain behaviour will lead to a desirable outcome, then they are more likely to have a positive attitude towards the behaviour. Alternatively, if individuals believe that a certain behaviour will lead to an undesirable or unfavorable outcome, then they are more likely to have a negative attitude towards the behaviour [13].

Illustrating the relationship between attitudes and food safety behaviour, Unklesbay et al. [14] developed an early survey instrument to assess the attitudes, practices, and knowledge of more than 800 college students. It was shown that women who had enrolled in a course including food safety information had significantly higher scores for both attitude and practice.

Relationships between knowledge, attitudes, and behaviour are more nuanced in this case because food safety attitudes might be crucial when it comes to food safety behaviour; international investigations using Structural Equation Model (SEM) may confirm this. Ko [15] used SEM to investigate the relationship between food safety knowledge, attitude, and Hazard Analysis Critical Control Points (HACCP) practices among 421 restaurant employees. It was similarly found that attitude mediated the relationship between knowledge and HACCP practices. Baser et al. [16] used SEM on food safety data regarding knowledge, attitude, and behaviour for 498 hotel staff and did not find any relationship between food safety knowledge and behaviour. However, the results from their analysis showed that there was a positive relationship between knowledge and attitude, as well as between attitude and behaviour. Furthermore, Sanlier and Baser [17] examined the relationships between food safety knowledge, attitudes, and self-reported behaviours using SEM. In a survey targeting 1219 young women, the mediating role of attitude was demonstrated. Results from that path analysis did not show any strong causal relationship between food safety knowledge and behaviour, while the results indicated a full mediation of the effect of knowledge on behaviour by attitude [17].

Limited data have been collected regarding the influence of different factors on food safety behaviours in Sweden, although these findings provide support for the hypothesis that the relationship between food safety knowledge and behaviour is mediated by attitude. These interactions are further investigated here. A Swedish survey performed on 606 university students has demonstrated that food safety education made a difference when it comes to food safety knowledge [5]. There were a significantly higher number of correct answers on a food safety knowledge questionnaire and this was correlated with more optimal self-reported food safety behavior. Thus, the investigation indicated a positive correlation between food safety education, knowledge and optimal self-reported food safety behaviours [5]. The present SEM study proposes to include the Background factor in order to possibly demonstrate a relationship with the factors of knowledge, attitude, and behaviour and to further evaluate the mediating role of attitude. This may have implications for the development of food safety education. International investigations using SEM have investigated the relationship between food safety knowledge, attitude, and behavior [15,16,17]. However, the novelty of this current investigation, as far as the authors are aware, is that there has not been an SEM analysis of food safety data which included the Background factor. In the present study, the Background factor includes the following issues: genus; experience cooking and handling different food items; experience taking a food safety course at different levels; and foremost source of food safety knowledge (Table 1). Specifically, the present study aims to investigate self-reported food safety background, knowledge, attitudes, and behaviour among university students in Sweden using SEM to examine factors influence on food safety behaviour.

## 2. Method

### 2.1. Design

Via a nationwide web-based questionnaire, food safety data on 606 students’ knowledge, attitudes, behaviours, and sources of food safety knowledge were collected. The questionnaire targeted students at 24 different universities in Sweden. The web-based questionnaire was distributed through social media, email, and various university contacts, and sent out as an open link via Google Forms [18]. The questionnaire, including demographic details and the process of collecting data, are as specified in Marklinder et al. [5].

For the present paper, an SEM analysis was applied to the data collected by the questionnaire, focusing on four factors: Background, Knowledge, Attitude, and Behaviour. The topic of the selected questions is presented in Table 1, Table 2, Table 3 and Table 4. These factors, which were constructed from variables with sufficient factor loadings (ranging from −1 to 1), were positioned in a predetermined structure (Figure 1). In this context the lowest factor is above 0.2 (or below −0.2). The analysis was carried out on a dataset with 408 observations and 61 variables, of which 30 were acceptable for use.

The structure was based on the questions concerning whether Background affects Knowledge, whether Knowledge affects Behaviour and Attitude, and whether Attitude affects Behaviour. Although modified in the present study, the structure has been confirmed to be valid in previous studies [15,16,17]. The model in Figure 1 was verified by a goodness of fit hypothesis test following the cut-off values prescribed by Hair et al. [19], the results of which can be seen in Table 5.

### 2.2. The Questionnaire

The construction of the questionnaire as well as the results from questions regarding demographics, food safety knowledge, and behaviour have been published in Marklinder et al. [5]. Question topics, including answer options and results regarding attitude, are presented in Appendix A.

### 2.3. Ethical Considerations

The present study was conducted according to the Swedish Research Council´s ethical guidelines, which are based on the Declaration of Helsinki [20]. The questionnaire was followed by a letter with information regarding the purpose of the investigation. Students were informed that their participation was voluntary and that they could withdraw from the study at any time. Participants were advised that questions must be answered individually and that we could assure them of total anonymity, as they would not be asked for personal data and answers would not be traceable to individuals. They were informed about the purpose of the questionnaire and that the results would be used in research. The first question in the questionnaire was as follows: “I have read the information above and agree to participate”. Answering “yes” to this question regarding participation made it possible to proceed with the questionnaire, which means that there was 100 percent confirmed consent among the respondents.

This kind of research does not require ethical vetting [21]. The present study does not handle any personal data relevant to Section 3 (Handling of Sensitive Data; Ethical Review of Research Involving Humans; Swedish Code of Statues, 2003:460;3-5§§).

## 3. Structural Equation Model

A structural equation model (SEM) is a tool for analysis of the interrelationships among latent variables measured using multiple correlated observable indications. The SEM in the present study was performed on the dataset in two steps. In the first step, a selection was made based on the relevance of the type of question to the objective of the analysis. Out of 606 respondents in the study by Marklinder et. al. [5], 408 were deemed usable for the SEM based on the respondent’s choice to opt out on certain questions in the questionnaire. The opt-outs were taken to include all observations from respondents who answered questions indecisively, such as “I don’t eat meat” or “I avoid this food for [a different reason]” or “I do not know” on any of the questions in the questionnaire. The original data contained many variables. Therefore, a selection of variables was undertaken according to determined criteria that were relevant for the analysis. To be sure of selecting them correctly it was performed in consultation with a food safety expert.

In the second step, factor loadings were tested to establish which variables could be used in the analysis. A factor loading is a standardized measure of the relationship between variables and the underlying structure, ranging from −1 to 1. A loading that is closer to 1 means that there is a strong effect between the variables and its factor or between factors. When forming factors in SEM, a loading from a variable to the latent variable is deemed acceptable at or above 0.5 [19]. However, according to Hair et al. [19], a loading of 0.3 is seen as sufficient to form the structure of a factor. Further, Matsunaga [22] pointed out that, as applied to social studies, loadings can be as low as 0.2. This cut-off point for factor loadings was applied to this study, along with the relevance of the variables themselves. More than half of the variables had loadings sufficient to their respective factors to be included, and all were relevant to the study. The theoretical underpinnings of the structure used for this analysis have been established by Ko [15], Baser et. al. [16] and Sanlier and Baser [17]. In addition, the Background factor was added to the present study. Figure 1 illustrates the four factors used for the model: Background; Knowledge; Attitude; and Behaviour.

### 3.1. The Factors

The different variables (B_1_–B_9_; K_1_–K_8_; A_1_–A_7_; H_1_–H_6_) forming the factors are explained in the Table 1, Table 2, Table 3 and Table 4. These are the variables that had the appropriate factor loadings to be included in the model. B_1_–B_9_ are variables for Background; K_1_–K_8_ are variables for Knowledge; A_1_–A_7_ are variables for Attitude; and H_1_–H_6_ are variables for Behaviour. In this study, the model path analysis has the novel addition of testing the causality of how a respondent’s Background affects Knowledge.

The questions used for Background were mainly concerned with whether respondents were women or men, had experience of cooking or handling different food items, the foremost source of their food safety knowledge, and whether it was informal (family and friends) or formal (food safety education) (Table 1). All variables were treated as binary, except for B_2_ and B_3_ which were on an ordinal scale.

The variables for Knowledge in the questionnaire, formed true/false questions with one or multiple answers, have been analyzed as binary data. These are dummy variables where the true answer to the question is valued as 1 and the rest are valued as 0 (Table 2).

The variables for Attitude included dealing with the importance of washing hands before food handling, after handling of raw minced meat, raw chicken, or raw eggs, and after toilet visits, as well as with cold food storage. One variable was an evaluation of the respondent’s level of food safety knowledge. The variables for Attitude were treated as ordinal. Response options were assessed as the means of six semantically different scales: “Very important”; “Rather important”; Neither important nor unimportant”; Not especially important”; Not at all important”; I have never been in this situation”/I never use leftovers” (Table 3).

The variables for Behaviour (H1–H6) were treated as ordinal variables where the least correct behaviour has the lowest score and the best behaviour has the highest value (Table 4).

The answer options for the questions in Table 1, Table 2, and Table 4 have been published in Marklinder et al. [5]. The answer options for the questions in Table 3 are provided in Appendix A.

### 3.2. Data Analysis

This structural equation model (Figure 1) is based upon ordinal data based on the nature of the questions in the questionnaire. As an example, in terms of attitude variables (Table 3), this ranges from 1, meaning not at all important, to 5, meaning very important. Behaviour variables (Table 4) are judged by what researchers deem to be the least desirable behaviour to the most desirable behaviour. As we use ordinal data, we used the diagonally weighted least squares (DWLS) estimator and polychoric correlations, in accordance with Yang-Wallentin et al. [23]. Binary variables such as dummy variables can be seen as a special case of ordinal variables. Because there is an intrinsic ordering in our study between how the respondents answered on binary questions DWLS can be used.

The collected data were processed and analysed in RStudio using the Lavaan package in order to perform the structural equation modelling. The Structural Equation Model’s goodness of fit was judged from four goodness of fit (GFI) sets; their indices are presented in Table 1. They consist of the comparative fit index (CFI), from comparison of the observed model to a null model; the absolute indices Root Mean Square Error of Approximation (RMSEA); the chi-square; and the normed chi-square (*χ*^2^ & *χ*^2^-normed). The chi-square test is usually used for testing the significant differences between observed data and estimated data. However, in SEM it is desirable to look for no differences between the data, i.e., a low observed *χ*^2^-statistic. Nevertheless, having performed a chi-square test, it is recommended to use *χ*^2^-normed as well as RMSEA, as both correct for chi-square inflation, along with CFI, which is the most widely used indice.

For this analysis, Hair et al. [19] recommend Goodness of Fit Index, as shown in Table 5, i.e., a *p*-value that is lower than 0.05, RMSEA lower than 0.08, and CFI larger than 0.90. These cut-off values are similar to those used by Ko [15], Baser et al. [16], and Sanlier and Baser [17].

## 4. Results

The analysis was carried out on a dataset with 408 observations and 61 variables, of which 30 were acceptable for use in the analysis (Figure 2) according to the two steps outlined in Section 3.

### 4.1. Factor Loadings

The Background variable B_5_: 0.98 (“Have you taken a course in food hygiene/safety and/or microbiology at university/college?”) was close to 1.0 (Table 1). Another strong factor loading was B_9_: 0.97 (“What is your foremost source of knowledge in what you know today about food hygiene?” (course at the university/college)) (Table 1). The lowest factor loadings in the present investigation were B_2_: 0.24 (“How often in the past year have you cooked for yourself or for somebody else in your household: food from raw ingredients such as minced meat, fish or chicken?”) and B_1_: 0.26 (“Is the respondent a woman?”). Other distinctive variables were B_6_: −0.40 (“Who is your foremost source of knowledge in what you know today about food hygiene?” (mother/female relative)) and B_7_: −0.31 (“Who is your foremost source of knowledge in what you know today about food hygiene?” (partner/friend)).

The Knowledge variable K_1_ had a loading of 0.81 regarding the correct answer (“Healthy people can carry the bacterium Staphylococcus aureus in their nose which may cause food poisoning” (true)), which is relatively high (Table 2). The variables K_7_, at 0.30 (“Proper refrigerated storage of food is one way to avoid food poisoning” (true)) and K_8_, at 0.36 (“What do you think is the optimal cooling temperature?” (4–5 ºC)), both had relatively low factor loadings.

Regarding Attitude, the variable A_3_: 0.81(“How important is it to wash your hands after handling raw chicken?”) was the highest (Table 3). There was a slightly lower loading for A_2_: 0.74 (“How important is it to wash your hands after handling raw minced meat?”). The variable A_5_: 0.54 (“How important is it to wash your hands after handling raw eggs?”) was the lowest among those three. Another distinctive factor loading regarding Attitude was the reference category A_7_: 0.79 (“How do you evaluate your level of food safety knowledge?”).

Regarding Behavior, the factor loadings for the variables H_1_–H_6_ varied between 0.26–0.48 (Table 4). The lowest variable was H_6_: 0.26 (“How do you clean your hands?”).

### 4.2. Results from the SEM

Results from the SEM are presented in Figure 2. Goodness of Fit indices indicated that the model had a good fit to the data, including hypothesis testing, with a significance of < 0.005. The regression coefficients are demonstrated in Figure 2: Background affected Knowledge (0.842). Knowledge had a strong direct effect on Attitude (0.605), and Attitude had a stronger influence on Behaviour (0.438) than on Knowledge (0.301). The regression coefficients that were closer to 1 indicated a stronger loading.

To verify that the model holds, the cut-off values for the goodness of fit indices introduced in Table 5 were compared with the results from the goodness of fit indices in Table 6. As can be seen, the *χ*^2^-test statistic is significantly lower than the null model, CFI was above 0.9, RMSEA was lower than 0.08, and *χ*^2^-normed was below 3.0. Hence, all goodness of fit indices were within their cut-off range (see Table 6), and the model fit proved to be good.

## 5. Discussion

### 5.1. Methodological Reflections

As an online questionnaire enabled the collection of data from 24 different universities in Sweden, all data were self-reported. Redmond and Griffith (2003_b_) [24] suggest that observational data provide the most reliable information denoting consumers’ actual food safety behaviour. According to da Cunha et al. [10], self-reported practices and observed practices are different, and should be used and discussed appropriately. However, in the present investigation, although it was based on self-reported data, we could see a direct effect of attitudes on behaviour.

The lowest factor loading in the present investigation were the variables B2: 0.24 (“How often in the past year have you cooked for yourself or for somebody else in your household: food from raw ingredients such as minced meat, fish or chicken?”); B1: 0.26 (“Is the respondent a woman?”) and H6: 0.26 (“How do you clean your hands?”). Low factor loadings indicate that these variables do not work very well as indicators for this particular factor. The higher the number, the more accurate a certain variable is in capturing information about the factor. As indicated earlier, Hair et al. [19] saw a loading of 0.3 as sufficient to form the structure of the factor, while for Matsunaga [22] this could be as low as 0.2 when applied to social studies. As the present study is framed as social science, using an online questionnaire and having a focus on knowledge, attitudes, and behaviour, the 0.2 loading cut-off is relevant. More than half of the variables were found to have sufficient loadings for their respective factors to be included, and all were relevant to the study (Figure 2). Gender was included because women and men may have different backgrounds in terms of spending time preparing foods in the private kitchen during their childhood and their upbringing [25].

Data for the present investigation were based on self-reporting of behavior; the regression coefficients from the SEM distinctly showed that Attitude had a stronger influence on Behaviour (0.438) than Knowledge (0.301). However, Knowledge had a strong direct effect on Attitude (0.605). According to the cut-off values for the goodness of fit indices introduced in Table 5, this model holds (Table 6). The results from the present investigation indicate that a background in food safety education and other sources of food safety awareness may develop food safety knowledge, which leads to a positive food safety attitude. This, in turn may have an impact on an optimal food safety behaviour.

### 5.2. Limitations

It should be emphasized that the data regarding food safety behaviour are self-reported, which increases the risk of bias. Several of the factor loadings in the present investigation were relatively low, especially the variables for Background B_1_ (0.26) and B_2_ (0.24) and the variable for behaviour H_6_ (0.26), which had signals that they had a low covariance with each other. Another limitation is that we used a convenience sample. The generalizability is limited to Swedish university students; however, they did represent 24 different universities in Sweden, ranging from north to south.

### 5.3. The Relations between the Background, Knowledge, Attitude, and Behaviour Factors

The present study clearly indicates that components in the Background factor involving having experience of any course in food safety, especially a university course, or having the foremost source of food safety knowledge be formal, i.e., food safety education at a university or college, had a high impact on the Knowledge factor; the regression coefficient was 0.842 (Figure 2). A possible reason for the relatively low factor loading on the variable B_1_ (“Is the respondent a woman?”) and B_2_ (“How often in the past year have you cooked for yourself or for somebody else in your household: food from raw ingredients such as minced meat, fish or chicken?”), at 0.26 and 0.24, respectively, could be explained by the fact that gender and the handling of raw food ingredients did not seem to be covariant with other variables within the same factor. Furthermore, the students may have acted in different ways because of this selection of food items. In Marklinder etal. [5], it was shown that knowledge differences between men and women were not statistically significant.

The loadings for the reference categories B_5_ (“Have you taken a course in food hygiene/safety and/or microbiology at university/college?”) and B_9_ (“What is your foremost source of knowledge in what you know today about food hygiene? “(course at the university/college)) compared to individuals whose foremost sources for food safety knowledge were B_6_ (“Who is your foremost source of knowledge in what you know today about food hygiene?” (mother/female relative)) or B_7_ (“Who is your foremost source of knowledge in what you know today about food hygiene?” (partner/friend)) are distinctively different from each other. Formal food safety education (B_5_ and B_9_) strongly reinforces Background as a positive influence on Knowledge. B_6_ and B_7_ show that if a partner or family member is seen as the primary knowledge source, this has an adverse effect on the Knowledge factor (Figure 2).

The Knowledge variables K_1_, K_7_, and K_8_ did not seem to be covariant with each other. The factor loading provides information regarding the variation of a certain variable, which has to do with correlations between the questions that belong to a certain factor. As far as the authors are aware, students´ knowledge regarding carrying of the pathogen Staphyloccus aureus in the noses of human beings, or knowledge about crucial cold food storage behaviour and correct refrigerator temperature, are not obviously related to each other.

The regression coefficient between Knowledge and Attitude was 0.605, and between food safety Knowledge and Behaviour was 0.301, clearly lower than between Attitude and Behaviour at 0.438.

The factor loading for the Behaviour variables H_1_–H_6_ ranged between 0.26–0.48 (Table 4) which in general was low. It is probable that these variables do not have a significantly stronger correlation with each other than with variables belonging to other factors. For these variables, the patterns are not clear.

### 5.4. Food Safety Attitudes

The high factor loading for the variable regarding the importance of washing hands after handling raw chicken may explain why there was little variation in the attitudes of students. Young consumers in Sweden are in general more often taught about and aware of the prevalence of Campylobacter in chicken than the prevalence of the shiga toxin-producing Escherichia coli bacteria (STEC) in minced meat [24]. Factor loadings for the variable regarding washing hands after handling raw chicken (A_3_) was higher than for handling minced meat (A_2_) (Table 3; Figure 2).

There was a low factor loading for handling of raw eggs (A_5_), a possible explanation for which may be because raw eggs never contain salmonella in Sweden [26]. Swedish students have no experience of salmonella in raw eggs [27]. In the Swedish food safety culture, consumers do not have to handle raw eggs with caution on this issue. In 1953, a severe Swedish salmonella epidemic involving almost 9000 bacteriologically-verified cases and at least 90 deaths was the starting point for the Swedish salmonella control programme [28]. Furthermore, meat from bovine animals and swine, poultry, meat, and eggs must have tested negative for salmonella in accordance with Commission Regulation (EC) No. 1688/2005 before export to Sweden [26]. Swedish salmonella control may mean that consumers think it is less important to wash their hands after handling raw eggs in comparison with raw chicken or raw minced meat. This must be seen as a particular national perspective, as in most other countries it would be correct to handle raw eggs with the same precautions.

### 5.5. Attitudes and Their Mediating Role

How might food safety education be developed to improve consumers´ food safety behavior? Ko [15], Baser et al. [16] and Sanlier and Baser [17] all discovered a near-zero regression coefficient between Knowledge and Behaviour, implying that knowledge has no effect on behaviour. As the regression coefficient between food safety Knowledge and Behaviour in the present study was 0.301 and clearly lower than between Attitude and Behaviour at 0.438, our results indicate that Attitude has a mediating role between Knowledge and Behaviour. These results are in accordance with those of Sanlier and Baser [15], who suggest that encouraging food safety attitudes rather than simply increasing the level of knowledge might be a more appropriate target for inducing behavioural change.

As attitudes are valued positively or negatively, this could explain how behaviours believed to offer more desirable consequences are favoured and negative attitudes are formed toward behaviours associated with mostly undesirable consequences. As an example, in order to decrease the risk for foodborne diseases, Zanin, et al. [6] suggested the importance of providing knowledge and encouraging appropriate attitudes through effective training to translate knowledge and attitudes into practices. The fostering of positive attitudes to food safety was emphasized. While food safety knowledge is important, food safety attitude has been shown to be a crucial factor.

The variables in the Background factor with the strongest loading were having taken a course in food hygiene/safety and/or microbiology at university/college, and the declaration that the foremost source of knowledge in terms of what the respondent knew about food hygiene today was from a course at university/college. In the directly opposite direction was the declaration that the foremost source of food safety knowledge was informal, i.e., family or friends. Logically, people’s background, their upbringing, and their formal education should affect their knowledge, which in turn affects their behaviour. Again, we make the point that this chain of latent variables needs to be added to attitude as a mediator of knowledge. Knowledge can in turn be seen as the mediator of background, and in this case, of what participants have learned from their formal education and upbringing. However, the topic remains complicated, as, for instance, the results from the multifaceted research regarding KAP have already shown [10,11]. We must distinguish consumers from food handlers in food companies. The most challenging difference is that food legislation is not applied in the private home. The challenge is to develop an educational methodology that contributes tools having an impact on food safety attitudes in order to facilitate optimal food safety behaviour change in private homes. Future SEM research might go one step further and focus on the causal relationships between background and attitude and between background and food safety behaviour.

## 6. Conclusions

The structural equation model for this study confirmed that the Background factor, involving topics such as having experience of food safety education or declaring that the foremost source of food safety knowledge was a formal university/college course, strongly influenced Knowledge. The Knowledge factor, in turn, strongly affected Attitude and did not directly affect Behaviour in the same way as Attitude. Thus, the Attitude factor seemed to have a mediating role between Knowledge and Behaviour. The hypothesis that the relationship between food safety knowledge and behavior is mediated by attitude was confirmed.

This study indicated that attitude has a stronger impact on behaviour than knowledge, which may have an impact on food safety behaviour. This has implications for the development of food safety education, and warrants further investigation and practical development.

## Figures and Tables

**Figure 1 foods-11-01595-f001:**
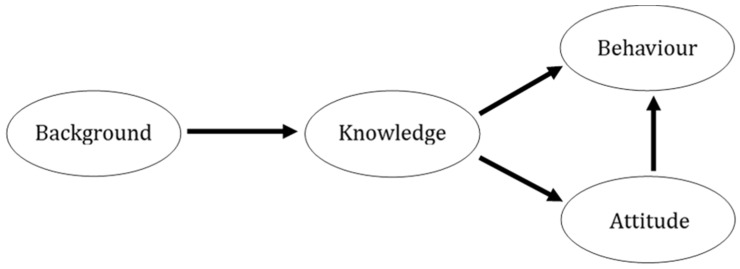
The model based upon the four factors, Background; Knowledge; Attitude; Behaviour, and 61 variables tested, of which 30 were acceptable for use.

**Figure 2 foods-11-01595-f002:**
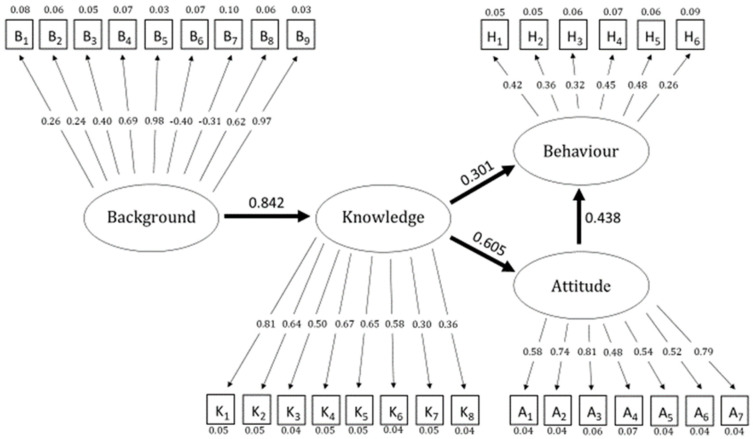
Structural Equation Model results are judged from the larger arrows, which are standardized regression coefficients between the factors Background, Knowledge, Attitude, and Behaviour, ranging from −1 to 1. Closer proximity to 1 or −1 indicates a strong positive or negative influence of one factor on another. Small arrows indicate the standardized factor loadings from variables that build up the underlying constructs (i.e., factors). Standard errors are expressed above each variable. The different variables (B_1_–B_9_; H_1_–H_6_; K_1_–K_8_; A_1_–A_7_) are explained in Table 1, Table 2, Table 3 and Table 4.

**Table 1 foods-11-01595-t001:** The variables for Background as included in the model (n = 408). Results of the questions regarding the topic background are published in Marklinder et al. [5]. The questions are freely translated from Swedish.

**B_1_**	Is the Respondent A Woman?
**B_2_**	How often in the past year have you cooked for yourself or for somebody else in your household: food from raw ingredients such as minced meat, fish or chicken?
**B_3_**	How often in the past year have you for yourself or for somebody else in your household: handled fresh vegetables/root vegetables/leeks/potatoes?
**B_4_**	Course in food hygiene/safety and/or microbiology at high school?
**B_5_**	Course in food hygiene/safety and/or microbiology at university/college?
**B_6_**	Foremost food safety source of knowledge? (mother/female relative)
**B_7_**	Foremost food safety source of knowledge? (partner/friend)
**B_8_**	Foremost food safety source of knowledge? (course at high school)
**B_9_**	Foremost food safety source of knowledge? (course at the university/college)

**Table 2 foods-11-01595-t002:** The variables for Knowledge as included in the model (n = 408). Results of the questions regarding the knowledge topic are published in Marklinder et al. [5]. The statements are freely translated from Swedish.

**K_1_**	True/False-Healthy people can carry the bacterium Staphylococcus aureus in their nose which may cause food poisoning (true)
**K_2_**	True/False-Eating a bloody/pink hamburger poses a risk of food poisoning. (true)
**K_3_**	True/False-Bacteria can grow in vacuum packaged products. (true)
**K_4_**	True/False-Listeria bacteria are mainly associated with raw chicken. (false)
**K_5_**	True/False-Foods heated to 54 °C are free of food poisoning bacteria. (false)
**K_6_**	True/False-You may risk food poisoning if you eat raw minced meat to test the seasoning. (true)
**K_7_**	True/False-Proper refrigerated storage of food is one way to avoid food poisoning. (true)
**K_8_**	1–2 °C/4–5 °C/7–8 °C-What do you think is the optimal cooling temperature? (4–5 °C).

**Table 3 foods-11-01595-t003:** The variables for Attitude as included in the model deal with food handling, toilet visits, and cold food storage in certain situations. Results for questions regarding the attitudes topic are shown in Appendix A. The questions are freely translated from Swedish.

**A_1_**	To wash your hands carefully before cooking food are for you?
**A_2_**	To wash your hands carefully after handling raw, minced meat are for you?
**A_3_**	To wash your hands carefully after handling raw chicken are for you?
**A_4_**	To wash your hands carefully after visiting the toilet are for you?
**A_5_**	To wash your hands carefully after handling raw eggs are for you?
**A_6_**	To cool leftovers within 4 h from cooking are for you?
**A_7_**	How do you evaluate your level of food safety knowledge?

**Table 4 foods-11-01595-t004:** The variables for Behaviour as included in the model dealing with certain behavioural situations regarding food safety (n = 408). Results for the questions regarding the behaviour topic are published in Marklinder et al. [5]. The questions are freely translated from Swedish.

**H_1_**	How do you know that the fried hamburger is properly cooked?
**H_2_**	How do you know that the chicken is properly cooked?
**H_3_**	You have cooked a large amount of food to be eaten later. How do you handle it after cooking?
**H_4_**	How often do you check the refrigerator temperature with a thermometer/thermo-element?
**H_5_**	You have cut raw meat and are now going to cut cucumber, tomato or lettuce, how do you do that?
**H_6_**	How do you clean your hands?

**Table 5 foods-11-01595-t005:** Goodness of Fit hypothesis test as judged from four sets of indices: the fit index (CFI), absolute indices Root Mean Square Error of Approximation (RMSEA), chi-square, and normed chi-square *χ*^2^ & *χ*^2^-normed).

**Goodness of Fit Index**		**Cut-Off Value**	
** *p* ** **-value**	P _obs_	**<**	0.05
** *χ* ** ** ^2^ **	*χ* ^2^ _obs_	**<**	estimator for the null-model
** *χ* ** ** ^2^ ** **-normed**	*χ*^2^_obs_/degrees of freedom	**<**	3:1 ratio
**Robust RMSEA**	RMSEA _obs_	**<**	0.08
**Robust CFI**	CFI _obs_	**>**	0.90

**Table 6 foods-11-01595-t006:** Goodness of Fit indices for the model and comparison with the null model.

Number of Free Parameters:	167
Goodness of Fit Index	Model	Null Model
*χ*^2^-teststatistic	1394.5	8155.6
Degrees of freedom	528	528
*p*-value	0.000	0.000
*χ*^2^-normed	2.840	
Robust RMSEA	0.067	
Robust CFI	0.901

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
