# Peer review of "A Structural Equation Model Demonstrating the Relationship between Food Safety Background, Knowledge, Attitudes and Behaviour among Swedish Students"

_foods, 2022, doi:10.3390/foods11111595_

Round 1
Reviewer 1 Report
The paper started with the novelty of the background, however, it was not detailed and discussed based on the literature. I recommend adjusting this point. Please find my detailed suggestions:
Abstract:
Lines 29 to 33 - Please provide the evidence of the effects, as you did on the influence of knowledge and the attitudes, in which you showed the number in brackets.
Introduction:
Lines 41 to 42 - Please provide a reference for the absence of doubt about the complex relationship between knowledge and behavior.
Lines 45 to 46 - It was demonstrated... by who? Please provide references and try to support your sentence with another study about TPB, there are many studies.
Line 48 - Please note that practices and behavior are not the same things, in this line you are using as synonymous.
Lines 48 to 54 - the terms presented in this paragraph are not connected. Please try to link practices with KAP with an optimistic bias, it is lacking connection between them.
Line 100 - As the background is not used in food safety studies, please provide the concept of the background.
What are the hypothesis of the study for the SEM?
Methods
Line 114 - What were the sufficient factors loadings?
section 2.2 - Just the attitudes questionnaire detailed? What about the other questionnaires? Please provide details, although it is already published.
Lines 147 and 148 - Please provide details of the criteria and the analysis determined in dialogue with a food safety expert. It is not clear what it means.
Line 157 - Please provide a reference for the acceptable value.
Results
Table 5 - Why does the total percentual not 100%?
Line 249 - Please provide an explanation about why the question about gender was important for the background?
Tables 6 and 7 - please provide notes with the legend of the acronymous. Note that all tables and figures need to be read and understood alone, without the text. What is d.f for instance?
Table 6 - Please translate P-värde from Sweeden to English.
Discussion
Line 289 - What do you mean inside brackets?
Lines 308 to 315 - The discussion is limited. What about the literature of KAP?
Conclusion - I suggest you do not conclude about food safety education as it was not assessed in the current study.
Author Response
Manuscript ID: foods-1715677 “A Structural Equation Model demonstrating the relationship between Food safety Background, Knowledge, Attitudes and Behaviour among Swedish students”.
Author reply: Dear reviewer, Thank you so much indeed for very useful comments. We have tried to make all corrections according to your requests.
1.
The paper started with the novelty of the background, however, it was not detailed and discussed based on the literature. I recommend adjusting this point. Please find my detailed suggestions:
Author reply: We would like to thank you for valuable comments on the manuscript and have tried to make corrections and improvements according to your wishes. Background has been presented and also discussed more in detail.
Abstract:Lines 29 to 33 - Please provide the evidence of the effects, as you did on the influence of knowledge and the attitudes, in which you showed the number in brackets.
Author reply: The lines 29-33 in the abstract have been revised and hopefully clarified: The regression coefficients that was demonstrated in the SEM for this study confirmed that the factor Background (genus; experience of handling different food items; experience of a food safety education course; the foremost sources of food safety knowledge) strongly influenced Knowledge (0.842). The factor Knowledge in turn, strongly affected Attitude (0.605) but it did not directly affect Behaviour (0.301) in the same way as Attitude. Thus, the factor Attitude seemed to have a mediating role between Knowledge and Behaviour. This study also indicated that students´ attitude on the importance of food safety may an impact on food safety behavior which should have implications for the development of food safety education.
Introduction:Lines 41 to 42 - Please provide a reference for the absence of doubt about the complex relationship between knowledge and behavior.
Author reply: We have added the reference: Da Cunha, D. Improving food safety practices in the food service industry. Curr. Opin. Food Sci. 2021, 42, 127-133
and the reference: Zanin, L.; da Cunha, D.; de Rosso, V.; Capriles, V.; Stedefeldt, E. Knowledge, attitudes and practices of food handlers in food safety: An integrative review. Food Res. Int. 2017, 100, 53-62.
And the new reference: Wilcock, A.; Pun, M.; Kanona, J.;Aung, M.Consumer attitudes, knowledge and behavior: a review of food safety issues. Trends in Food Sci. & Techn.2004,15,56-66.
Lines 45 to 46 - It was demonstrated... by who? Please provide references and try to support your sentence with another study about TPB, there are many studies.
Author reply: It was Mullan et al who demonstrated that ones other factors were taken into accounts it was shown that knowledge was not the sole predictor of behavior.
However, we also added the reference Sanlier & Baser (2019) who also demonstrated the mediating role by attitude in a SEM: Sanlier, N.; Baser, F. The relationship among Food safety knowledge, attitude and behaviour of young Turkish women. J. Am. Coll. Nutr. 2019, 39, 224-234.
Line 48 - Please note that practices and behavior are not the same things, in this line you are using as synonymous.
Author reply: We have removed the words: “Food safety behavior, or..” and fully understand the difference between behaviours and practices. However, as the KAP is still interesting we keep the information regarding KAP.
Lines 48 to 54 - the terms presented in this paragraph are not connected. Please try to link practices with KAP with an optimistic bias, it is lacking connection between them.
Author reply: we added the references(Da Cunha, 2021; Mullan, 2010).
A clarification regarding riskrelated optimismHuman beings are complex and factors such as optimistic bias, lack of motivation or inadequate infrastructure may hamper an optimal behavior [Da Cunca; Mullan, 2010]. Further, riskperception is a complicated concept because people in general are more likely to believe that they will win the highest profit rather than that they will suffer from something negative such as for instance food poisoning. The “optimistic bias” or “riskrelated optimism”, defined by Weinstein, 1984 [ ], may prevent consumers from absorbing information and following rules.
The reference below has been added.
Weinstein, N. D. (1984). Why it won't happen to me: Perceptions of risk factors and susceptibility. Health Psychology, 3(5), 431–457. https://doi.org/10.1037/0278-6133.3.5.431
Line 100 - As the background is not used in food safety studies, please provide the concept of the background.
The author reply: We have made a clarification regarding the factor Background: However, the novelty of this current investigation, as far as the authors are aware, is that there has not been a SEM analysis on food safety data which included the factor Background. In present study the factor Background includes the following issues: genus; the experience of handling different food items; experience of taking a food safety course on different levels; the foremost source of food safety knowledge [Table 1].
What are the hypothesis of the study for the SEM?
Author reply: One hypothes: The relationship between food safety knowledge and behavior is mediated by attitudes
Further, the model in itself included a hypothesis testing with a significance of < 0.005 (Table 5 and 6)
Methods
Line 114 - What were the sufficient factors loadings?
Author reply: The factors loadings are presented above Figure 1.
section 2.2 - Just the attitudes questionnaire detailed? What about the other questionnaires? Please provide details, although it is already published.
Author reply: The Tables 1-4 have been tidied up and the text around them has been clarified. The performance and results for Table 1, 2 and 4 can be found in Marklinder et al (2020). The results for Table 3 have been put in Supplemental S1.
Lines 147 and 148 - Please provide details of the criteria and the analysis determined in dialogue with a food safety expert. It is not clear what it means.
Author reply: We have corrected the sentence which hopefully will make it more clearly: “Their factor loadings were selected according to determined criteria that were relevant for the analysis. To be sure of the having selected them correctly it was performed in dialogue with a food safety expert. All observations from respondents who answered questions indecisively such as “I don’t eat meat” or “I avoid this food for [a different reason]” or “I do not know” were removed from the dataset”, because they were not relevant for the SEM.
Line 157 - Please provide a reference for the acceptable value.
Author reply:We have included the reference Hair et al., (2014)
Results
Table 5 - Why does the total percentual not 100%?
Author reply: The row has been deleted due to the request from another reviewer. The table regarding attitude and results is put to the Supplemental Table S1.
Line 249 - Please provide an explanation about why the question about gender was important for the background?
Author reply: Genus was included because women and men may have different background in spending time preparing foods in the private kitchen during their childhood. We added this sentence ad also a new reference: (Lange et al. 2018).
Lange, M.; Göranzon, H.; Fleig, L.; Marklinder, I. Adolescents´sources for food safety knowledge and trust. Br. Food.J.2018, 120, 549-562.
Tables 6 and 7 - please provide notes with the legend of the acronymous. Note that all tables and figures need to be read and understood alone, without the text. What is d.f for instance? The table has been developed and d.f= “degrees of freedom” is completed.
Table 6 - Please translate P-värde from Swedish to English.
Author reply: This is all developed and also translated correctly!
Discussion
Line 289 - What do you mean inside brackets?
Author reply: This was easy corrected! Should be 2003b
Lines 308 to 315 - The discussion is limited. What about the literature of KAP?
Author reply: Yes, I added some more about KAP: “However, the topic is overall complicated as for instance the results from the multifaceted research regarding KAP already has been shown [da Cunha et al 2019; da Cunha 2021]. We must also distinguish the consumers from the food handlers in food companies. The most challenging is that food legislation will not be applied in a private home. The challenge is to develop an educational methodology that contributes tools having an impact on food safety attitudes in order to facilitate optimal food safety behaviour change in the private homes. Future SEM research might go one step further and focus on the causal relationship between background and attitude, and between background and food safety behavior”.
Conclusion - I suggest you do not conclude about food safety education as it was not assessed in the current study.
Author reply: The sentence “We conclude that formal food safety education is important” is deleted.
Reviewer 2 Report
This study evaluating the influence of background on Swedish students’ KAP possibly provided insights for Sweden. The following are specific suggestions to improve the manuscript.
General comment: The manuscript needs considerable improvement in writing quality and organization of information
Line 112: What is meant by “some of the data”? Needs clarification. Is there a secondary analysis of previously collected data?
Line 117: For the figure 1 caption, what is meant by “off-set”?
Line 123 to 129: Were only the two questions provided, or are these examples of questions? Why are only the questions related to attitude presented? What about the background, knowledge, and behavior questions? Maybe refer the reader to Tables 1 to 4 for the questions/items for each construct.
Line 132 to 134: The sentence implies that consent was obtained after the participants were given the questionnaire. Is this correct? Seems unusual.
Line 146: Suggest more detail about the “opt out on certain questions” exclusion criteria. I think those details may be in lines 148 to 151, but a sentence in-between makes it unclear.
Tables 1 – 4: Suggest listing all items used to measure the constructs along with the means or frequencies (for dichotomous variables) and initial factor loadings. Designate which items were dropped (or kept) for further analysis. Also, the formatting needs to be corrected for the table. The response scale for all questions/items needs to be added as footnotes.
Table 5: Why is data for attitudes the only construct presented? The last row of the table seems unnecessary as the frequencies were based on 408 participants and 408 participants answered all of the questions.
Line 316: Suggest that the use of a convenience sample also be added as a limitation. Also, generalizability is limited to Swedish college students (I assume there are no demographics provided).
Line 325: “Gender” is not a source of food safety knowledge.
General comment: Suggest adding a table with participant demographics and a summary of the information in the text.
General comment: Were the items/constructs used in this study based on prior studies? If yes, what was the translation process to Swedish? (Assuming respondents were presented with the questionnaire in Swedish).
General comment: The discussion could be strengthened by comparing KAP results to other studies.
Author Response
Manuscript ID: foods-1715677 “A Structural Equation Model demonstrating the relationship between Food safety Background, Knowledge, Attitudes and Behaviour among Swedish students”.
This study evaluating the influence of background on Swedish students’ KAP possibly provided insights for Sweden. The following are specific suggestions to improve the manuscript.
General comment: The manuscript needs considerable improvement in writing quality and organization of information.
Author reply: Dear reviewer, Thank you so much indeed for very useful comments. We have tried to make all corrections according to your requests.
Line 112: What is meant by “some of the data”? Needs clarification. Is there a secondary analysis of previously collected data?
Author reply: We have changed the meaning of the sentences and highlighted that the selected questions used are specified in Table 1-4.
Line 117: For the figure 1 caption, what is meant by “off-set”?
Author reply: The word off-set is removed.
Line 123 to 129: Were only the two questions provided, or are these examples of questions? Why are only the questions related to attitude presented? What about the background, knowledge, and behavior questions? Maybe refer the reader to Tables 1 to 4 for the questions/items for each construct.
Author reply: Improvements are done and we also refer to Tables 1-4.
Line 132 to 134: The sentence implies that consent was obtained after the participants were given the questionnaire. Is this correct? Seems unusual.
Author reply: The text has been developed and especially regarding the obtained consent as follows:
“The first question in the questionnaire was as follows: “ I have taken part of the information above and do agree to participate”. Just if answering “yes” on this question regarding participation made was possible to proceed which means that there was 100 percent confirmed consent among the participant respondents.”
Line 146: Suggest more detail about the “opt out on certain questions” exclusion criteria. I think those details may be in lines 148 to 151, but a sentence in-between makes it unclear.
Author reply: Improvements have been done in the text!
Tables 1 – 4: Suggest listing all items used to measure the constructs along with the means or frequencies (for dichotomous variables) and initial factor loadings. Designate which items were dropped (or kept) for further analysis. Also, the formatting needs to be corrected for the table. The response scale for all questions/items needs to be added as footnotes.
Author reply: The Tables 1-4 have been tidied up and the text around them has been clarified. The performance and results for Table 1, 2 and 4 can be found in Marklinder et al., (2020). The results for Table 3 has been put in Supplemental S1.
Table 5: Why is data for attitudes the only construct presented? The last row of the table seems unnecessary as the frequencies were based on 408 participants and 408 participants answered all of the questions.
Author reply: The last row has been deleted. The results from the questions regarding Background, Knowledge and behavior were published in Marklinder et al., (2020) which is also mentioned before. The attitude results have been put in Supoplemental Table S1.
Line 316: Suggest that the use of a convenience sample also be added as a limitation. Also, generalizability is limited to Swedish college students (I assume there are no demographics provided).
Author reply: This has been added!
Line 325: “Gender” is not a source of food safety knowledge.
Author reply: Right, it is not meant to be a source of knowledge but is connected to the factor Background. Genus was included because women and men may have different background in spending time preparing foods in the private kitchen during their childhood. We added this sentence and also a new reference: (Lange et al. 2018).
Lange, M.; Göranzon, H.; Fleig, L.; Marklinder, I. Adolescents´sources for food safety knowledge and trust. Br. Food.J.2018, 120, 549-562.
General comment: Suggest adding a table with participant demographics and a summary of the information in the text.
Author reply: All demographics are published in Marklinder et al., 2020.
General comment: Were the items/constructs used in this study based on prior studies? If yes, what was the translation process to Swedish? (Assuming respondents were presented with the questionnaire in Swedish).
Author reply: Yes, the questionnaire was performed in Swedish language. Before submitting the manuscript the translation was spell- and grammar tested.
General comment: The discussion could be strengthened by comparing KAP results to other studies.
Author reply: Yes, some more about KAP has been added to the discussion: “However, the topic is overall complicated as for instance the results from the multifaceted research regarding KAP already has been shown [da Cunha et al 2019; da Cunha 2021]. We must also distinguish the consumers from the food handlers in food companies. The most challenging is that food legislation will not be applied in a private home. The challenge is to develop an educational methodology that contributes tools having an impact on food safety attitudes in order to facilitate optimal food safety behaviour change in the private homes. Future SEM research might go one step further and focus on the causal relationship between background and attitude, and between background and food safety behavior”.
Reviewer 3 Report
The paper gives the impression of having been drafted in a great hurry, without paying due attention and care to its presentation: there are typing errors, repetitions (the fact that the Background factor was added to the original model is repeated too many times), tables named in the wrong place/order (first table 6 is mentioned, then table 5), some brackets are missing, and above all, it is not tidy: Many considerations included within the discussion section should be moved to the concluding considerations; also the whole limitations section); the Attitude factor is discussed separately from the other factors; it is not clear how some variables are measured/treated.
I enclose the pdf file with some comments that highlight the disorder that characterises the work.
Good luck

Author Response
Manuscript ID: foods-1715677 “A Structural Equation Model demonstrating the relationship between Food safety Background, Knowledge, Attitudes and Behaviour among Swedish students”.
Submission Date 21 April 2022
Author reply: Dear reviewer, Thank you so much indeed for very useful comments. We have tried to make all corrections according to your requests.
The paper gives the impression of having been drafted in a great hurry, without paying due attention and care to its presentation: there are typing errors, repetitions (the fact that the Background factor was added to the original model is repeated too many times).
Author reply: We have gone through the manuscript and hope it has been improved. Some errors have been removed.
tables named in the wrong place/order (first table 6 is mentioned, then table 5),
Author reply: The table 5 has been organized correctly.
some brackets are missing, and above all, it is not tidy:
Author reply: This was easy corrected! Should be 2003b
Many considerations included within the discussion section should be moved to the concluding considerations; also the whole limitations section);
the Attitude factor is discussed separately from the other factors; it is not clear how some variables are measured/treated.
Author reply: The Tables 1-4 have been tidied up and the text around them has been clarified. The performance and results for Table 1, 2 and 4 can be found in Marklinder et al., (2020). The results for Table 3 have been put in Supplemental Table S1. The last row has been removed due to your request.
Author reply: The 2.2. regarding the questionnaire has now gone through a major revision. All questions have been described and the title now is just: “The questionnaire”. However, as the results from the background, demographics, knowledge questions and behavior questions are already published, in present study the results from the attitude questionnaire are presented. We could refer to Marklinder et al., 2020 for more details.
I enclose the pdf file with some comments that highlight the disorder that characterize the work.
Author reply: The manuscript is over all tidied up and corrected according to your wishes. Hopefully it will be fully changed and improved now! Thank you so much for very good input!
Round 2
Reviewer 1 Report
The article has been improved substantially. I just recommend a final grammar check and a check on the references numbers.
Author Response
REVIEWER NUMBER 1Round 2
Manuscript ID: foods-1715677 “A Structural Equation Model demonstrating the relationship between Food safety Background, Knowledge, Attitudes and Behaviour among Swedish students”.
Thank you so much indeed for very useful comments. We have tried to make all corrections according to your requests.
Reviewer 1: The article has been improved substantially. I just recommend a final grammar check and a check on the references numbers.
Author reply: We have performed a spell/grammar check of the manuscript.
We have also corrected the text references and also in the reference list. We have removed some (year) in connection to some references and marked all those places with yellow. As for instance Marklinder et al. (2020) [5] has been changed to Marklinder et al. [5].

Reviewer 2 Report
While improvements have been made, the manuscript still needs considerable improvement in writing quality and organization of information. There are multiple spelling, grammar, and sentence structure problems that need correcting. Suggest using a professional editor.
For tables 1 – 4, minimally need to clarify what is meant by “results” (i.e., this word is used in the table descriptions, but what results are you referring to? Maybe add a footnote to address this).
A supplemental table (S1) was referenced in the manuscript but was not included.
Author Response
REVIEWER 2, Round 2
Manuscript ID: foods-1715677 “A Structural Equation Model demonstrating the relationship between Food safety Background, Knowledge, Attitudes and Behaviour among Swedish students”.
Dear Reviewer, Thank you so much indeed for very useful comments. We have tried to make all corrections according to your requests.
Comments from Reviewer 2: While improvements have been made, the manuscript still needs considerable improvement in writing quality and organization of information. There are multiple spelling, grammar, and sentence structure problems that need correcting. Suggest using a professional editor.
Author reply: The manuscript has been spell/grammar checked. Also, some mistakes in the references have been corrected.
Comments from Reviewer 2: For tables 1 – 4, minimally need to clarify what is meant by “results” (i.e., this word is used in the table descriptions, but what results are you referring to? Maybe add a footnote to address this).
Author reply: The table texts have been developed by adding respectively topic regarding the results e.g. in Table 1: “Results of the questions regarding the topic background are published in Marklinder et al. [5].”
A supplemental table (S1) was referenced in the manuscript but was not included.
Author reply: Sorry for if the supplemental table did not appear as it should. Now we have submitted the Supplemental Table (S1).

Reviewer 3 Report
Dear authors,
thank you for your improvements to the paper. There are still some of my previous comments disregarded as the position of the tables or the improvement of the conclusion section. Please, try to satisfy my requests or justify your choice not to opt into them.

Author Response
Reviewer 3
Manuscript ID: foods-1715677 “A Structural Equation Model demonstrating the relationship between Food safety Background, Knowledge, Attitudes and Behaviour among Swedish students”.
Author reply: Dear reviewer, Thank you so much for very useful comments. We are sorry for that in the first round we missed your valuable information that was put in the speech bubbles in the attached document. However, now we have incorporated them. Overall, we have tried to make all corrections according to your requests.
Reviewers´ comments: Thank you for your improvements to the paper. There are still some of my previous comments disregarded as the position of the tables or the improvement of the conclusion section. Please, try to satisfy my requests or justify your choice not to opt into them.
Author reply: Thank you so much for your valuable comments. We are sorry for that we missed some of your comments communicated in the attached document. By now we have tried to go through them carefully. The manuscript has been spell/grammar checked again and some references have been corrected (yellow marking). Further, Tables 1-4 have been tidied up and the text is no longer aligned to left. Tables 5 and 6 has been re-organized. The brackets and quotation mark that were messy have also been re-organized. Some repetitive sentences have been removed. The table text for Tables 1-4 have been changed and we have moved some text from the caption to the main text, according to your wishes.
Reviewers´ comment: Many considerations included within the discussion section should be moved to the concluding considerations; also the whole limitations section).
Author reply: Regarding the section for limitations (5.2) we prefer to keep it in within the discussion part tightly connected to Methodological considerations. It may be a tradition, but if this is OK, we would prefer to keep it at the 5.2. Hopefully, it would be satisfactory?
Author reply: Also, regarding the concluding remarks - “The challenge is to develop an educational methodology that contributes tools having an impact on food safety attitudes in order to facilitate optimal food safety behaviour change in the private homes. Future SEM research might go one step further and focus on the causal relationship between background and attitude, and between background and food safety behavior” - we would prefer to keep these sentences in the discussion part as this is not really a conclusion, but just a recommendation for further work.
Author reply: Regarding the communication via the speech bubbles in the attached document:
- 2 Questionnaire is now corrected. The text below has been developed and comprises not only the attitude questions but also the questions regarding Background, statements of knowledge and questions regarding behaviour. It will be a lot better now; thank you so much!
- The “61 variables text” has been developed and also a space has been added: “The analysis was carried out on a dataset with 408 observations and 61 variables of which 30 were acceptable to be used”.
- Reviewers comments: “It sound slightly different from what is reported in the caption of the table 1. Furthermore, how it is measured/treated the variable B1? Of course not binary, but how?”
- Author reply: The text has been removed from the caption of the table 1, and is further developed in the main text: “The questions used for Background were mainly concerned with whether respondents were women or men, had experience of cooking or handling different food items, their foremost source from where they obtained their food safety knowledge, and whether it was informal (family and friends) or formal (food safety education) (Table 1). All variables were treated as binary, except B2 and B3 which are on an ordinal scale”.
- Reviewers comments: “Add a call for table 2”. Author reply: A call for Table 2, For Table 3 and Table 4 have now been done (yellow marking).
- 5. The coefficients Knowledge and Behaviour has been corrected (yellow marking).
- Regarding section 5.3: “The impact of Background” has been changed to “The relations between the factors Background, Knowledge, Attitude and Behaviour”. Thank you so much for these comments as it was not just about Background but also about the factor Knowledge, Attitude and Behaviour. The part related to Knowledge was also moved just to organize the text (yellow marking).
- Also, the result information regarding Attitudes has been moved to section 5.3. according to your recommendation. (Yellow marking): “The regression coefficient between Knowledge and Attitude was 0.605 and, between food safety Knowledge and Behaviour, was 0.301 and clearly lower than between Attitude and Behaviour at 0.438”.
We do hope you will accept that we keep the caption “Attitudes and their mediating role” as we have been eager to focus on the role of attitudes in this manuscript and wanted to highlight it to combine it also with the text and argument in the Introduction.
Author reply: Thank you so much for your patience and valuable input!
